



# Identification, monitoring, and reaction kinetics of reactive trace species using time-resolved mid-infrared quantum cascade laser absorption spectroscopy: Development, characterisation, and initial results for the Criegee intermediate CH2OO

Zara S. Mir[1], Matthew Jamieson[1], Nicholas R. Greenall[1], Paul W. Seakins[1], Mark A. Blitz[1,2], Daniel Stone[1]

[1]School of Chemistry, University of Leeds, Leeds, LS2 9JT, UK
[2]National Centre for Atmospheric Science, University of Leeds, Leeds, LS2 9JT, UK

*Correspondence to*: Daniel Stone (d.stone@leeds.ac.uk)

**Abstract.** The chemistry and reaction kinetics of reactive species dominate changes to the composition of complex chemical systems, including Earth's atmosphere. Laboratory experiments to identify reactive species and their reaction products, and to monitor their reaction kinetics and product yields, are key to our understanding of complex systems. In this work we describe the development and characterisation of an experiment using laser flash photolysis coupled with time-resolved mid-infrared (mid-IR) quantum cascade laser (QCL) absorption spectroscopy, with initial results reported for measurements of the infrared

spectrum, kinetics, and product yields for the reaction of the $CH_2OO$ Criegee intermediate with $SO_2$. The instrument presented has high spectral ($< 0.004$ cm$^{-1}$) and temporal ($< 5$ µs) resolution, and is able to monitor kinetics with a dynamic range to at least 20,000 s$^{-1}$. Results obtained at 298 K and pressures between 20 and 100 Torr gave a rate coefficient for the reaction of $CH_2OO$ with $SO_2$ of $(3.83 \pm 0.63) \times 10^{-11}$ cm$^3$ s$^{-1}$, which compares well to the current IUPAC recommendation of $\left(3.70^{+0.45}_{-0.40}\right) \times 10^{-11}$ cm$^3$ s$^{-1}$. A limit of detection of $4.0 \times 10^{-5}$, in absorbance terms, can be achieved, which equates to a limit of detection

of $\sim 2 \times 10^{11}$ cm$^{-3}$ for $CH_2OO$, monitored at 1285.7 cm$^{-1}$, based on the detection pathlength of $(218 \pm 20)$ cm. Initial results, directly monitoring $SO_3$ at 1388.7 cm$^{-1}$, demonstrate that $SO_3$ is the reaction product for $CH_2OO + SO_2$. The use of mid-IR QCL absorption spectroscopy offers significant advantages over alternative techniques commonly used to determine reaction kinetics, such as laser-induced fluorescence (LIF) or ultraviolet absorption spectroscopy owing to the greater number of species to which IR measurements can be applied. There are also significant advantages over alternative IR techniques, such as step-

scan FT-IR, owing to the coherence and increased intensity and spectral resolution of the QCL source, and in terms of cost. The instrument described in this work has potential applications in atmospheric chemistry, astrochemistry, combustion chemistry, and in the monitoring of trace species in industrial processes and medical diagnostics.





# 1 Introduction

The behaviour of reactive intermediates is critical to understanding the chemistry of complex systems. In the gas phase, reactive
intermediates govern the chemistry and composition of planetary atmospheres (Blitz and Seakins, 2012), the interstellar
medium and star-forming regions (Herbst, 2001), as well as controlling combustion processes and autoignition (Pilling et al.,
1995; Zador et al., 2011). In the Earth's atmosphere, the chemistry of reactive intermediates determines the rate at which
compounds emitted into the atmosphere are removed and transformed into other species, and thus drives changes to air quality
and climate (Monks, 2005; Von Schneidemesser et al., 2015).


Experimental investigation of the spectroscopy, kinetics, and reaction mechanisms of reactive intermediates is key to
understanding the behaviour of such species, requiring sensitive and specific detection techniques capable of monitoring
changes in concentrations during the course of reactions commonly occurring on microsecond to millisecond timescales. Flash
photolysis experiments, in which a reactive species is generated rapidly by a brief pulse of light from a flashlamp or, more
commonly, a laser and then monitored throughout its subsequent reactions, have enabled the study of many reactions of
interest. However, while the flash photolysis method can be coupled to a wide range of techniques to determine the kinetics of
a reaction, limitations remain, particularly surrounding the identification of reaction products and measurements of product
yields (Seakins, 2007).

For the study of fast reactions, spectroscopic or mass spectrometric techniques are typically required to provide the necessary
time resolution. Laser-induced fluorescence (LIF) spectroscopy has been demonstrated to have both high sensitivity and
specificity for the measurement and identification of reactants and products, but is not an absolute technique and so requires
either calibration or the use of an internal standard to determine yields (Carr et al., 2007). Moreover, LIF can only be applied
to the relatively small number of species that exhibit fluorescence spectra. Mass spectrometry can be applied more widely,
particularly if soft ionisation techniques such as photoionisation are employed, but such techniques can be costly and require
sampling of a reaction mixture into an ionisation and detection region which can limit investigations to low pressure regimes
(Fockenberg et al., 1999; Blitz et al., 2007; Osborn et al., 2008; Middaugh et al., 2018). Absorption techniques can therefore
be beneficial as these can provide absolute measurements over a wide range of temperatures and pressures.

Absorption measurements based on ultraviolet (UV) spectroscopy have been used successfully to measure the kinetics of a
broad variety of reactions, with advantages for radical-radical reactions where absolute concentrations are required. While UV
absorption spectra can be relatively broad and featureless (Orlando and Tyndall, 2012), developments in the use of broadband
light sources have enabled the separation of multiple absorbing species with overlapping spectra (Cossel et al., 2017),
particularly if any of the spectra display distinctive vibronic structure. Multipass (Lewis et al., 2018) and cavity enhanced
techniques (Cossel et al., 2017) can also offer significant improvements to sensitivity. However, UV absorption experiments



are typically employed to monitor changes in reactant concentrations to determine reaction kinetics, but identification of products and measurement of product yields is often not possible owing to a lack of suitable UV absorption features or low UV absorption cross-sections for product species. There is thus interest in the development and use of infrared (IR) techniques which, despite lower absorption cross-sections compared to the UV, have the potential to be implemented more extensively

since most species exhibit some features in the IR region of the spectrum, with IR spectroscopy offering the potential for structural determination and unique identification of reactants and products. Infrared absorption techniques are also often advantageous over other spectroscopic methods, particularly at relatively low temperatures and pressure, since IR transitions are typically not dissociative (Taatjes and Hershberger, 2001; Hodgkinson and Tatam, 2013), and as a result do not suffer lifetime broadening or probe-induced photochemistry. Additionally, Doppler broadening in the IR is less problematic than in

the UV or visible, leading to better resolution of closely spaced features (Taatjes and Hershberger, 2001; Hodgkinson and Tatam, 2013).

For many reactions of atmospheric interest, products have been determined by long-path Fourier transform infrared (FT-IR) absorption spectroscopy in atmospheric simulation chambers used to study reactions at a mechanistic level (Doussin et al.,

1997; Glowacki et al., 2007; Nilsson et al., 2009; Seakins, 2010), but this method has relatively poor time resolution compared to direct studies of elementary reactions. Such studies can be influenced by secondary chemistry and wall reactions which may transform reactive products into more stable species on the timescale of the experiment. Step-scan FT-IR experiments (Su et al., 2013; Huang et al., 2007), in which spectra are recorded at successive time points during the course of a reaction, offer improved time resolution and aid direct identification of reaction products. However, step-scan FT-IR has relatively poor

sensitivity and spectral resolution compared to other techniques and is not typically employed to investigate reaction kinetics owing to the length of time required to record a suitable time profile.

Diode lasers (Taatjes and Hershberger, 2001), quantum cascade lasers (QCLs) (Faist et al., 1994; Hofstetter and Faist, 2003; Yao et al., 2012; Zhang et al., 2014; Pecharroman-Gallego, 2013), and frequency comb lasers (Fleisher et al., 2014; Bjork et

al., 2016; Roberts et al., 2020) can be used to give both high spectral resolution and high temporal resolution, and can be applied to measurements of transient species. Frequency comb lasers can provide extremely high resolution spectra, but there are, at present, relatively few examples (Bjork et al., 2016) of their application in chemistry and chemical kinetics owing to relatively high cost and complexity of the experimental setup. In contrast to mid-IR lead salt diode lasers, mid-infrared QCLs do not require cryogenic operation and do not suffer issues related to mode-hopping, uneven diode quality, or unpredictable

tunability (Hodgkinson and Tatam, 2013). The spectral output of QCLs can be tailored to transitions over a wide range, providing access to regions of the spectrum not readily accessible with diode lasers or frequency comb lasers (Shahmohammadi et al., 2019; Yao et al., 2012; Zhang et al., 2014). QCLs have high output powers, and can reach several hundred mW to ~1 W (Hodgkinson and Tatam, 2013; Pecharroman-Gallego, 2013; Yao et al., 2012; Zhang et al., 2014), compared to typical powers of a few mW for diode lasers (Hodgkinson and Tatam, 2013), and absorbances on the order of $10^{-6}$ have been reported for



QCL-based infrared absorption experiments for stable species (Borri et al., 2006). The high resolution, specificity, and sensitivity of QCL-based absorption techniques has led to the development of a number of field instruments for detection of trace species in the atmosphere (Li et al., 2013; Du et al., 2019), including $CH_4$ (Mcmanus et al., 2010; Kostinek et al., 2019), $CO_2$ (Kostinek et al., 2019), OCS (Mcmanus et al., 2010), $N_2O$ (Mcmanus et al., 2010; Banik et al., 2017; Kostinek et al., 2019), $H_2O$ (Mcmanus et al., 2010), HCOOH (Herndon et al., 2007), and HONO (Cui et al., 2019) which can be challenging

to monitor by other methods. The high power and spectral resolution of QCL light sources, enabling sensitive and specific experiments in regions of the spectrum characterised by strong fundamental transitions, have also led to interest from the chemical kinetics community.

QCLs consist of layers of semiconductor material that create a series of coupled quantum wells in which the layer thickness

determines the depth of the quantum well and thus the energy of emitted photons (Yao et al., 2012; Faist et al., 1994). In contrast to diode lasers, which involve electronic transitions between conduction and valence bands, laser action in QCLs involves intersub-band transitions within the conduction band, typically via a three-level system (Yao et al., 2012; Gmachl et al., 2001; Curl et al., 2010). In the absence of an electric field, electrons are confined in the quantum wells within an injector region. When an electric field is applied, the quantum wells within the injector region align and electrons are injected into an

upper intersub-band state, creating a population inversion with an intermediate level. Relaxation of electrons to a lower intersub-band state, resulting in photon emission, followed by rapid tunnelling of electrons from the lower intersub-band state into the injector region of the next layer creates a cascade of electrons as the layer structure is traversed, leading to increased photon emission and significant optical gain (Yao et al., 2012). Broadband emission can be achieved using external cavity (EC) QCLs, which can give coverage of several hundred $cm^{-1}$, or Fabry-Perót (FP) QCLs, giving coverage of ~50 $cm^{-1}$, while

single mode emission relevant to this work is achieved using distributed feedback (DFB) QCLs (Shahmohammadi et al., 2019). For emission in the mid-IR, DFB QCLs can be operated at room temperature, with control of the temperature of the semiconductor and the applied current allowing fine tuning of the laser output within a range of ~5 $cm^{-1}$ (Shahmohammadi et al., 2019).

Pulsed QCLs have been used to measure the production of CO in the combustion of *n*-heptane (Nasir and Farooq, 2019), and have been used to determine the IR absorption spectrum and cross-sections of the Criegee intermediate $CH_2OO$ (Chang et al., 2017; Chang et al., 2018b; Luo et al., 2018a). $CH_2OO$ is a reactive species produced in the atmosphere during the ozonolysis of unsaturated volatile organic compounds (VOCs) that has been of recent interest as a result of developments in photolytic sources (Welz et al., 2012) for detailed laboratory studies which have revealed a more significant role in atmospheric chemistry

than previously expected (Chhantyal-Pun et al., 2020; Percival et al., 2013). Quasi-continuous QCLs, pulsed QCLs in which the pulse period is relatively long compared to the lifetime of the species under investigation, have also been used to investigate the spectra and kinetics of $CH_2OO$ (Chang et al., 2018a), and cw QCLs have been used to investigate the kinetics of $CH_2OO$ (Luo et al., 2018b; Luo et al., 2019; Li et al., 2019; Li et al., 2020) and other larger Criegee intermediates (Luo et al., 2018b),





as well as the spectroscopy and kinetics of the atmospherically important peroxy radicals $HO_2$ (Miyano and Tonokura, 2011;
Sakamoto and Tonokura, 2012) and $CH_3O_2$ (Chattopadhyay et al., 2018). While mid-IR QCLs have been employed to study
the kinetics and spectroscopy of reactive species relevant to atmospheric chemistry, there are still few examples of the use of
QCL-based techniques to identify reaction products and to determine product yields.

In this work, we report the development, characterisation, and initial results from a robust and economical experiment using
laser flash photolysis coupled with time-resolved mid-infrared QCL absorption spectroscopy that can be applied to a wide
range of problems in atmospheric chemistry and beyond. We describe the experimental setup (Sect. 2), time-averaged
measurements of absorption spectra of stable species (Sect. 3), and time-resolved measurements of photolytically generated
species and reaction products (Sect. 4) that can be used to determine spectra of reactive species as well as reaction kinetics and
product yields. Factors affecting the limit of detection are also discussed (Sect. 5).

## 2 Experimental

A schematic of the experimental setup is given in Figure 1. The reaction cell has been used previously in IR diode laser
experiments (Qian et al., 2000, 2001; Choi et al., 2006) and consisted of a central cylindrical stainless steel cell of length 70
cm and internal diameter 40 mm, connected via bellows at each end to stainless steel sections of length 15 cm and 7 cm internal
diameter which are furnished with 60 mm diameter UV grade $CaF_2$ windows (Crystran).


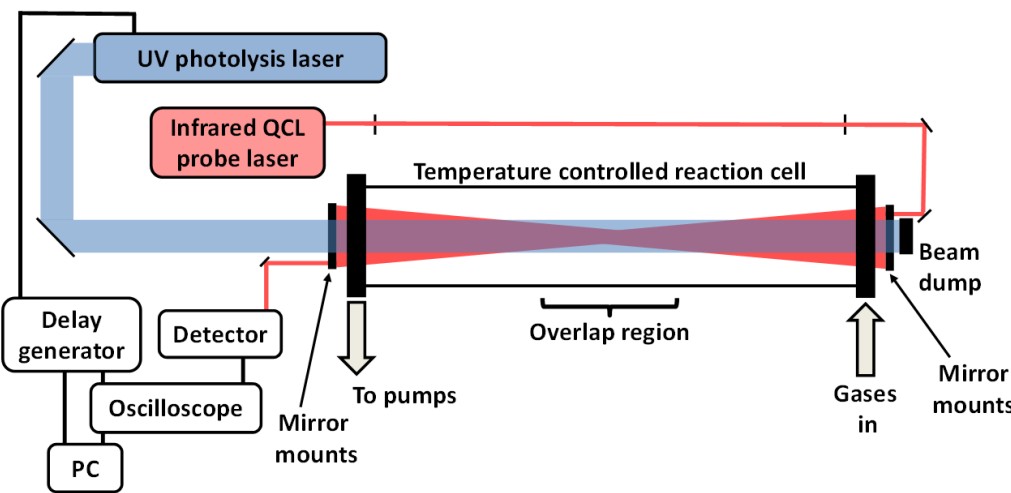

Figure 1: Schematic of the experimental apparatus. The probe beam passes through the reaction cell multiple times, with mirrors
in a circular arrangement on the mounts positioned at each end of the cell such that the probe passes from the top of the cell to the
bottom or from left to right in any individual pass through the cell, resulting in the overlap between the photolysis and probe beams
occurring in the centre of the cell.



Reagent gases, such as NO (BOC Special Gases, 99.5 %) and $SO_2$ (Sigma-Aldrich, 99.9%) (Sect. 4), were prepared manometrically at known concentrations in $N_2$ and stored in glass bulbs before mixing with $N_2$ (BOC, 99.99 %), and in some cases, $O_2$ (BOC, 99.999 %), in a stainless steel mixing line at known flow rates determined by calibrated mass flow controllers

(Tylan, 2900 series). For experiments involving species in the liquid phase at room temperature, such as $CH_3I$ (Sigma-Aldrich, 99 %) and $CH_2I_2$ (Sigma-Aldrich, 99 %) (Sect. 3 and 4), the liquid was contained within a bubbler held at 0 °C and entrained into the vapour phase by passing a known flow of $N_2$ through the bubbler which was subsequently combined with the main gas flow in the mixing line. Gas mixtures prepared in the mixing line were passed into the reaction cell via one of the end sections, with the other end section connected to a rotary pump (Edwards E2M28). The total pressure in the cell was monitored

by a capacitance manometer (MKS Instruments, 626A) and controlled by throttling the exit of the cell to the pump, with cell pressures from < 5 Torr to above atmospheric pressure achievable. Experiments reported in this work were performed at room temperature, although temperature control of the reaction cell is possible by surrounding the cell with ceramic heaters (Watlow, WATROD tubular heater), a bath filled with an appropriate solvent (e.g. methanol) chilled by a refrigerated immersion probe (LabPlant Refrigerated Immersion Probe, RP-100CD), or a dry ice slush bath (Choi et al., 2006), all of which are available in

this laboratory.

For experiments involving reactive species generated by photolysis, chemistry within the cell was initiated by the fourth harmonic of an Nd:YAG laser (Continuum Powerlite 8010), giving 266 nm (typical fluence 30 mJ cm$^{-2}$). The photolysis beam has diameter ~ 1 cm and was aligned through the centre of the reaction cell using a pair of mirrors (ThorLabs NB1-K04). For

all experiments reported in this work, the repetition rate of the photolysis laser was set to 1 Hz and the flow rate through the reaction cell was sufficiently high to ensure that a fresh gas mixture was photolysed for each photolysis shot.

Infrared probe radiation was provided by one of two cw mid-IR DFB QCLs, depending on the application, providing radiation at wavenumbers of ~1286 cm$^{-1}$ (~7.77 µm) (Alpes Lasers) and ~1390 cm$^{-1}$ (~7.19 µm) (ThorLabs) with a tuning range of ~ 5

cm$^{-1}$ around each centre. Temperature and current control of the QCL, which determines the precise output wavenumber of a given QCL, were controlled by a combined laser current and thermoelectric (TEC) controller (ThorLabs, ITC4002QCL). The current and TEC controller was operated in constant current mode, which provides current control up to 2 A in 0.1 mA steps with accuracy ±(0.1 % + 800 µA) and stability < 150 µA, and temperature control between 123 and 423 K in 0.001 K steps with stability < 0.002 K.


QCLs were housed in high heat load (HHL) packages, which were each integrated with a ZnSe aspheric lens to collimate the output beam (divergence < 6 mrad), and mounted on a heatsink (Hamamatsu HHL mount, A117909-1) to dissipate excess heat. Mounted QCLs were positioned on a pitch and yaw stage (ThorLabs, PY003/M) on an adjustable height platform (ThorLabs, C1519/M) to aid alignment through two adjustable aperture irises (ThorLabs, ID8/M) which were separated by >





70 cm. A collimated laser diode operating at 635 nm (ThorLabs, CPS635R) was co-aligned through the irises via a mirror (ThorLabs, PF10-03-G01) on a flip-mount (ThorLabs, FM90/M) to guide alignment of the QCL beam through the reaction cell. A pair of 1" Al mirrors (ThorLabs, PF10-03-G01) situated after the second iris were used to direct the beam into the reaction cell, with the beam focussed into the centre of the cell by a 1" ZnSe plano-convex lens (focal length 1000 mm, ThorLabs, LA7753-G).

The probe IR beam was aligned through the cell in a multipass arrangement to increase the total pathlength and sensitivity, with the mirrors reflecting the probe beam external to the cell. Custom built mirror mounts with a central 12 mm hole were located exterior to the cell and were used to mount six Ag mirrors (12 mm diameter, 2.4 m radius of curvature, Knight Optical) in a circular arrangement at each end of the cell, similarly to the arrangement previously described for the multipass UV absorption experiment in this laboratory (Lewis et al., 2018). These mirrors can be aligned independently of each other via three alignment screws for each mirror, and enable up to 13 passes through the reaction cell. On the final pass of the IR probe beam through the cell the beam was directed onto a 1" Au off-axis parabolic mirror (reflected focal length 4", ThorLabs, MPD149-M01) and focussed onto the detector. The detector was a DC-coupled photovoltaic Mercury-Cadmium-Telluride (MCT) detector (Vigo System, PVMI-3TE-8-2x2-TO8-wBaF2-35), integrated with a pre-amplifier (Vigo System, MIP-DC-100k-F-M4) with bandwidth up to 100 kHz.

The signal was transferred to an oscilloscope via a sheathed BNC cable for data collection and processing, with the settings on the oscilloscope dictating the time resolution of the experiment. For measurements of stable species, all spectra reported in this work were recorded using a traditional oscilloscope (LeCroy Waverunner-2, LT262, 350 MHz, 1 GS/s sample rate, 8 bit resolution), while for reactive species produced in photolytic experiments, the use of a PicoScope (Pico Technology, PicoScope 6402C, 250 MHz, 5 GS/s sample rate, 12 bit resolution) was also investigated. Data acquisition is discussed further in Sect. 4 and 5.

Synchronisation of the photolysis laser and oscilloscope was achieved using a custom-built digital delay generator based on National Instruments hardware, with the overall experiment and data collection controlled by custom LabVIEW software. In addition to measurements of the probe intensity as a function of time at a fixed QCL output wavenumber, the software enabled stepping of the current applied to the QCL at a set temperature to vary the output wavenumber, which is used to measure the variation in the probe intensity across the tuning range with and without a sample present to determine the spectra of stable species. For reactive species produced by, or following, photolysis, the average pre- and post-photolysis intensities were determined over specified time ranges at each current setting to give the absorption spectrum. Further details are given in Sect. 4.



## 3 Time averaged experiments

Characterisation of the output wavenumber of the QCL was achieved through measurement of the absorption spectra of stable
species with well-defined rovibrational spectra. Spectra for $SO_2$ and $CH_3I$ were recorded by measuring the variation in QCL intensity across the tuning range of the QCL for the cell filled with $N_2$ and for the cell filled with a mixture of $N_2$ and the species of interest under otherwise identical conditions. The absorbance, $A_{\tilde{v}}$, at each wavenumber $\tilde{v}$, was calculated from the Beer-Lambert law (Equation 1):

$$A_{\tilde{v}} = \ln\left(\frac{I_{\tilde{v},0}}{I_{\tilde{v}}}\right) = \sigma_{\tilde{v}}[C]l \qquad\qquad \text{(Equation 1)}$$

where $I_{\tilde{v},0}$ and $I_{\tilde{v}}$ are the intensities at wavenumber $\tilde{v}$ without and with the species of interest present, respectively, $\sigma_{\tilde{v}}$ is the absorption cross-section at wavenumber $\tilde{v}$, [C] is the concentration, and $l$ is the path length of the IR probe beam through the sample. In measurements of stable species in experiments in which no photolysis takes place, the total path length can be determined from the geometry of the cell and optical arrangement, and was ~13 m for the measurements discussed here.

Figure 2 shows the normalised observed absorbance spectrum for $SO_2$ alongside the comparison to normalised spectrum available on the HITRAN database (Gordon et al., 2017; Kochanov et al., 2019) to illustrate the calibration of the QCL output wavenumber with the current. The measured spectrum indicates the capacity for high resolution measurements made possible by the narrow laser linewidth.





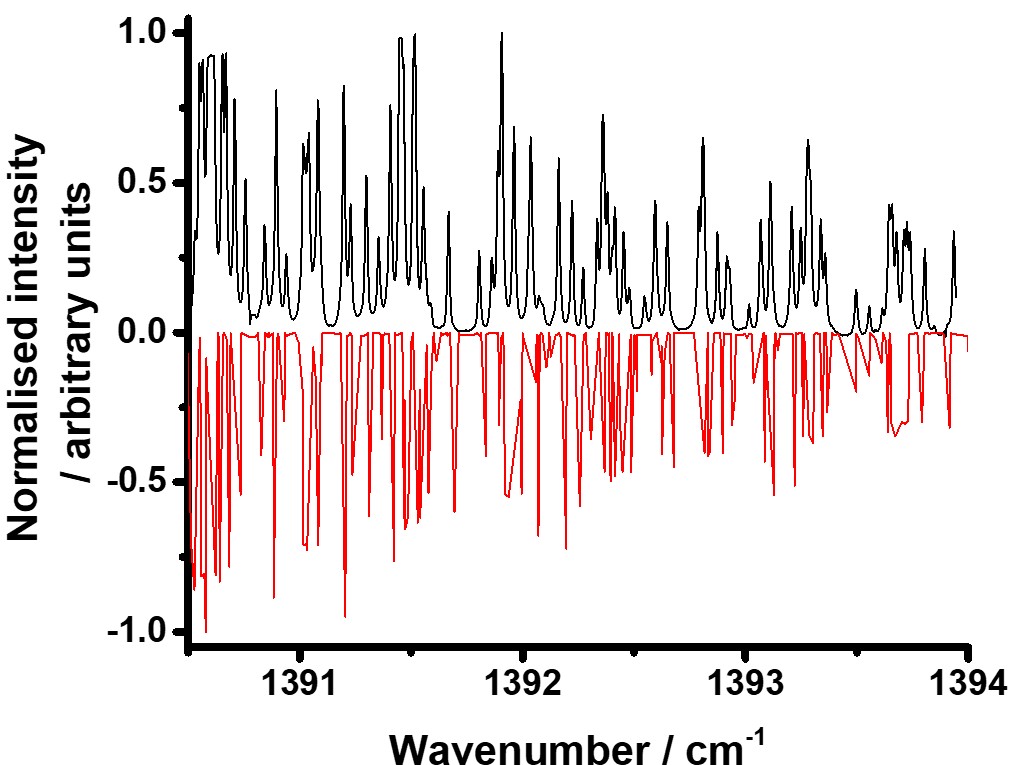

Figure 2: Normalised observed spectra (black) for SO₂ with the corresponding normalised spectrum reported on the HITRAN database (Gordon et al., 2017; Kochanov et al., 2019) (multiplied by -1) (red). HITRAN data are given as line intensities, and thus do not account for any line broadening.

## 4 Time resolved experiments

### 4.1 Stable species

Photolytic experiments were initially performed for a system demonstrating a step-change in the time-resolved signal intensity at a given wavenumber on photolysis, such that photolysis led to the removal of the species under investigation with no significant further chemistry on the timescale of the measurements. Figure 3 shows the time-resolved absorbance observed on 266 nm photolysis of $CH_3I/N_2$ mixtures and the average post-photolysis change in absorbance as a function of the initial $CH_3I$ concentration. Photolysis of $CH_3I$ leads to a decrease in concentration, and thus an increase in signal intensity and a negative absorbance. The extent of change in absorbance reflects the absorption cross-section at the measurement wavenumber, the change in concentration, and the effective path length resulting from the overlap between the UV photolysis beam and the IR probe beam. For the 266 nm laser fluence of 30 mJ cm⁻² and $CH_3I$ absorption cross-section of $9.7 \times 10^{-19}$ cm² at the photolysis



wavelength (IUPAC) (Atkinson et al., 2008), a change in $CH_3I$ concentration of 4 % is expected on photolysis. For the absorbance data shown in Figure 3 and an estimated infrared cross-section of $CH_3$ at ~1287 $cm^{-1}$ of $2 \times 10^{-21}$ $cm^2$ (HITRAN)

(Gordon et al., 2017; Kochanov et al., 2019), the effective path length of the IR probe beam for these measurements can thus be estimated as $(290 \pm 30)$ cm. The effective path length is discussed further in Sect. 4.2.

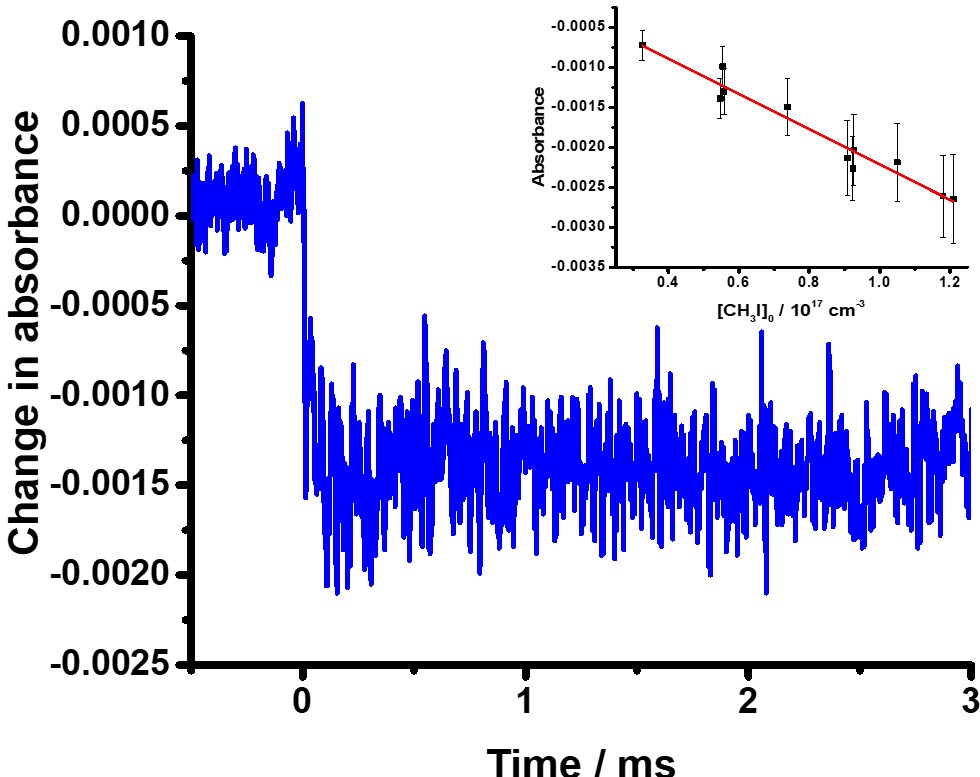

**Figure 3: Changes in absorbance at ~1287 $cm^{-1}$ observed on 266 nm photolysis of $CH_3I$. For the data shown in the main figure $[CH_3I]_0 = 5.5 \times 10^{16}$ $cm^{-3}$, $p$ = 50 Torr, and laser fluence 30 mJ $cm^{-2}$. The data shown are the means of 200 photolysis shots. The inset**
**shows the average post-photolysis (0.5 ms < $t$ < 3 ms) change in absorbance as a function of the initial $CH_3I$ concentration (black points) with the line of best fit (red).**

## 4.2 Reactive species

The behaviour of reactive species was investigated through 266 nm photolysis of $CH_2I_2/O_2/N_2$ and $CH_2I_2/O_2/N_2/SO_2$ mixtures

(Welz et al., 2012), resulting in the rapid production ($k_{CH2I+O2}[O_2] > 2 \times 10^5$ $s^{-1}$) of the Criegee intermediate $CH_2OO$ (R1-R2), followed by its removal through the $CH_2OO$ self-reaction (R3), $CH_2OO + I$ (R4), and, in the presence of $SO_2$, $CH_2OO + SO_2$ (R5):





$$\begin{array}{llll}
CH_2I_2 + h\nu & \rightarrow & CH_2I + I & (R1) \\
CH_2I + O_2 & \rightarrow & CH_2OO + I & (R2a) \\
& \rightarrow & CH_2IO_2 & (R2b) \\
CH_2OO + CH_2OO & \rightarrow & 2\ HCHO + O_2 & (R3) \\
CH_2OO + I & \rightarrow & CH_2IO_2 & (R4) \\
CH_2OO + SO_2 & \rightarrow & HCHO + SO_3 & (R5)
\end{array}$$

Figure 4 shows the spectrum obtained in the absence of $SO_2$ by measuring the average pre- and post-photolysis absorbances for an observed time-profile for a given QCL current setting, and thus a given wavenumber, and then stepping to the next current setting and repeating. The step in current for these experiments was 0.1 mA, the smallest step-size available with the current controller used, giving steps of $< 0.002$ cm$^{-1}$ across the range investigated. The pre-photolysis region was defined as -4000 μs to -500 μs, owing to detection of some radiofrequency noise associated with the Q-switch delay of the photolysis laser

which was set to 280 μs, and the post-photolysis region as 500 μs to 6000 μs, where $t = 0$ is the time at which the photolysis laser is fired. Experiments were performed at 50 Torr and each time-resolved trace was averaged for 1000 photolysis shots. The concentration of $CH_2I_2$ for these experiments, determined from the flow rates of gases in the cell, the vapour pressure of $CH_2I_2$ and measurements of the saturation of the flow with $CH_2I_2$ in previous experiments with similar flow rates, was ~2 × $10^{14}$ cm$^{-3}$, and the 266 nm laser fluence was 30 mJ cm$^{-2}$, giving an expected initial $CH_2OO$ concentration of ~7 × $10^{12}$ cm$^{-3}$

using previous measurements of the yield of R2a (Stone et al., 2013). The spectrum measured in this work is in good agreement with that reported previously ( Chang et al., 2017; Chang et al., 2018a). The resolution observed in this work is similar to the resolution of $< 0.004$ cm$^{-1}$ reported in previous work (Chang et al., 2017) based on the observation of non-overlapped peaks, although the linewidth of the QCLs used in this work and in that in the previous measurement of the spectrum (Chang et al., 2018a) should enable higher resolution of $< 0.002$ cm$^{-1}$ when observing species with more closely spaced features.

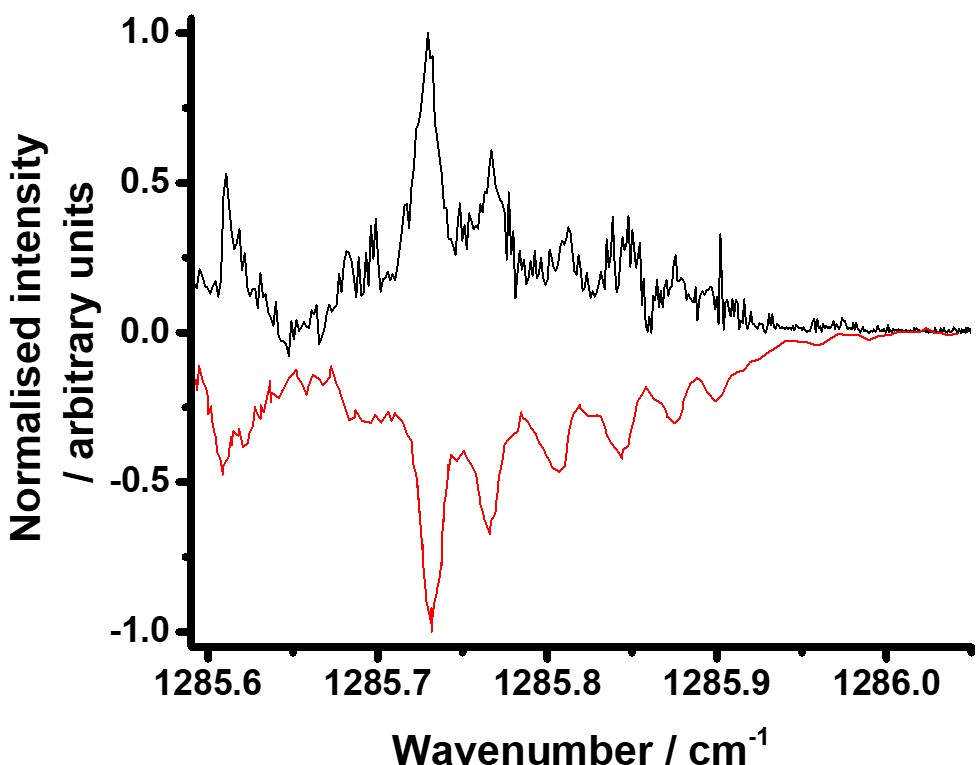

**Figure 4: Normalised CH₂OO spectrum obtained in this work (black) and that reported previously (Chang et al., 2018a) (multiplied by -1) (red). Measurements made in this work were obtained at a total pressure of 50 Torr at 298 K in the absence of SO₂, with [CH₂I₂] ~2 × 10¹⁴ cm⁻³ and a laser fluence of 30 mJ cm⁻². The data shown are the means of 1000 photolysis shots at each current setting on the QCL with the wavenumber calibrated using the results reported in previous work (Chang et al., 2018a).**

Kinetics describing the loss of $CH_2OO$ were monitored in separate experiments with the QCL set at the peak in the $CH_2OO$ spectrum (~1285.73 cm⁻¹). Figure 5 shows an example $CH_2OO$ decay obtained at a pressure of 50 Torr in the absence of $SO_2$. The data shown in Figure 5 indicate the ability to tune the QCL to a particular spectral feature, and for the QCL to remain tuned to that feature for prolonged periods of time during which kinetics experiments, and repeat measurements, can be performed without any drift in spectral position. QCLs thus offer significant advantages for kinetics experiments over alternative mid-IR sources such as lead salt diode lasers which can suffer from mode-hopping, uneven diode quality, and unpredictable tuneability (Hodgkinson and Tatam, 2013). Such behaviour of alternative mid-IR sources can require sophisticated techniques to scan over a spectral feature, identify the peak, and then wait until the laser remains stable for a sufficient period of time to perform the desired experiment (Qian et al., 2000).


In the absence of any additional co-reactant such as $SO_2$, the loss of $CH_2OO$ (as shown in Figure 5) is dominated by reactions R3 and R4 (Mir et al., 2020). Since R3 and R4 are not first-order, the kinetics describing the loss of $CH_2OO$ are dependent on absolute $CH_2OO$ concentrations. The kinetics of R3 and R4 have been determined in our previous work (Mir et al., 2020), and the absorption cross-section of $CH_2OO$ has been measured (Chang et al., 2018b) to be $(3.9 \pm 0.6) \times 10^{-18}$ cm² at the peak of its

spectrum at ~1285.73 cm⁻¹ at a total pressure of 50 Torr. The numerical integration package FACSIMILE (MCPA Software, 2014) was therefore used to fit to the measured absorbances to determine the physical losses of $CH_2OO$ owing to diffusion out of the probe beam (approximated by a first-order rate coefficient $k_{phys}$) and the effective path length ($l$) of the probe beam required to convert the absorbances to concentrations compatible with the measured $CH_2OO$ absorption cross-section of $(3.9 \pm 0.6) \times 10^{-18}$ cm² (Chang et al., 2018b) and the known kinetics for the mechanism used in the model (Table 1). Fits were

performed for experiments in which precursor concentrations were varied in the range $0.1 - 5.0 \times 10^{14}$ cm⁻³. Initial concentrations of $CH_2I$, iodine atoms, and $CH_2IO_2$ were calculated in the model relative to those for $CH_2OO$ using the yields of R1 and R2 determined in our previous work (Stone et al., 2013).

| Reaction | Rate coefficient, $k^a$ / cm³ s⁻¹ or $k^b$ / s⁻¹ | Reference |
|---|---|---|
| $CH_2I + O_2 \rightarrow CH_2OO + I$ | $\beta \times 1.5 \times 10^{-12}$ | Stone et al., 2013; Masaki et al., 1995; Eskola et al., 2006 |
| $CH_2I + O_2 \rightarrow CH_2IO_2$ | $(1-\beta) \times 1.5 \times 10^{-12}$ | Stone et al., 2013; Masaki et al., 1995; Eskola et al., 2006 |
| $CH_2OO + CH_2OO \rightarrow 2\ HCHO + O_2$ | $8.0 \times 10^{-11}$ | Mir et al., 2020 |
| $CH_2OO + I \rightarrow$ products | $3.4 \times 10^{-11}$ | Mir et al., 2020 |
| $CH_2OO \rightarrow$ loss | $k_{phys}$ | Determined in fit |
| $CH_2IO_2 + CH_2IO_2 \rightarrow 2\ CH_2IO + O_2$ | $9.0 \times 10^{-11}$ | Gravestock et al., 2010 |
| $CH_2IO_2 + I \rightarrow CH_2IO + IO$ | $3.5 \times 10^{-11}$ | Gravestock et al., 2010 |
| $CH_2IO \rightarrow HCHO + I$ | $1 \times 10^5$ | Gravestock et al., 2010 |

**Table 1: Reactions and rate coefficients used in the model to fit to experimental observations of $CH_2OO$ absorbances to determine**
**the effective path length of the probe beam. The absorbances at time zero, $k_{phys}$ (describing physical losses of $CH_2OO$ such as diffusion) and the effective path length were fit in the model. Initial concentrations of $CH_2I$, iodine atoms, and $CH_2IO_2$ were determined relative to the initial concentrations determined for $CH_2OO$ using the results of our previous measurements of the yields of $CH_2I + O_2$, where $\beta$ indicates the yield of $CH_2OO$ from $CH_2I + O_2$ (Stone et al., 2013). The rate coefficient $k_{phys}$ represents physical losses of $CH_2OO$ such as diffusion out of the probe region, and was determined in the fits to this work. ªSecond-order reactions,**
**ᵇfirst-order reactions.**

The fits to the measured absorbances gave $k_{phys} = (504 \pm 40)$ s⁻¹ and an effective path length of $(218 \pm 20)$ cm. Figure 5 shows an example fit to the data. While the effective path length determined from experiments involving $CH_2OO$ $((218 \pm 20)$ cm) is lower than that determined from experiments involving $CH_3I$ $(290 \pm 30)$ cm, Sect. 4.1), the results are in broad agreement and

the system was realigned between experiments which may explain the difference.



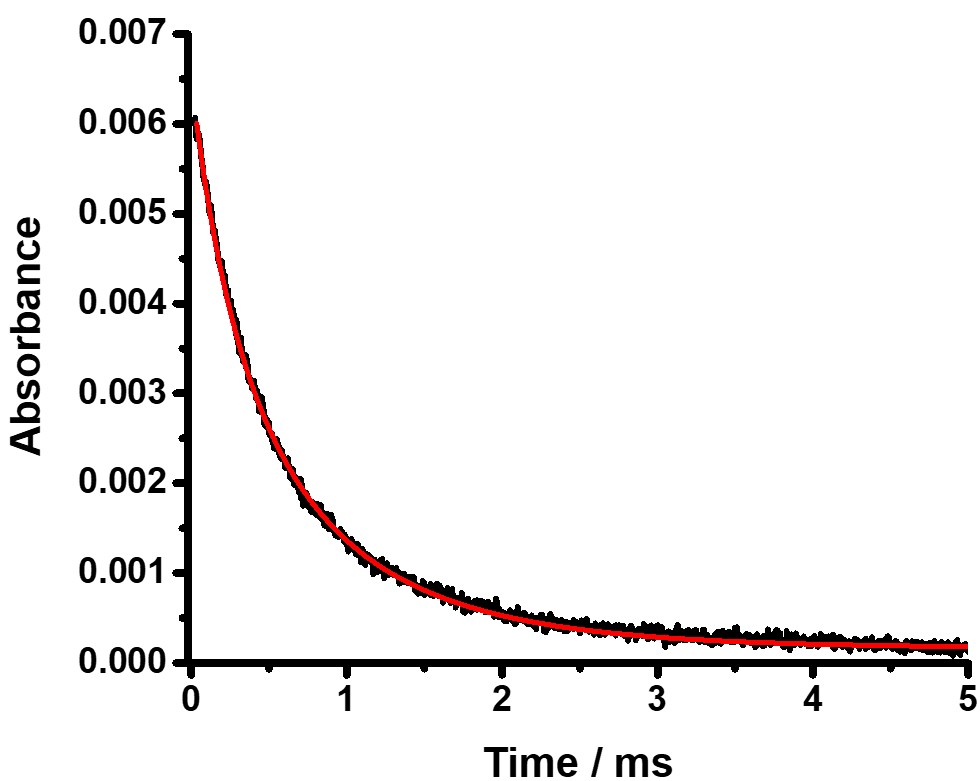

**Figure 5: Time-resolved absorbance for CH$_2$OO obtained at a wavenumber of ~1285.73 cm$^{-1}$. Data are the average of 1000 photolysis shots at a total pressure of 50 Torr. For these data the [CH$_2$I$_2$] = 1.6 × 10$^{14}$ cm$^{-3}$ and [SO$_2$] = 0. The fit to the data using the model containing the reactions given in Table 1 is shown by the solid red line, which gave $k_{phys}$ = (504 ± 40) s$^{-1}$ and $l$ = (218 ± 20) cm.**

In the presence of excess SO$_2$ the observed decays of CH$_2$OO are dominated by R5 and the loss of CH$_2$OO can be described by pseudo-first-order kinetics (Equation 2).

$$A_t = A_{t=0} \exp(-k't) \qquad \text{(Equation 2)}$$

where $A_t$ is the absorbance at time $t$, $A_0$ is the absorbance at time zero, and $k'$ is the pseudo-first-order rate coefficient, given by $k' = k_5[SO_2] + k_{phys}$ where $k_5$ is the bimolecular rate coefficient for reaction between CH$_2$OO and SO$_2$ and $k_{phys}$ is the rate coefficient representing physical losses of CH$_2$OO such as diffusion out of the probe region.

Fits of Equation 2 to the observed CH$_2$OO decays obtained in the presence of SO$_2$ were performed to determine the pseudo-first-order rate coefficients, $k'$, which can plotted against the known SO$_2$ concentration (Sect. 2) to give the bimolecular rate coefficient $k_5$. Figure 6 shows typical bimolecular plots of $k'$ against the SO$_2$ concentration, giving $k_5$ = (3.73 ± 0.19) × 10$^{-11}$ cm$^3$ s$^{-1}$ at 20 Torr, $k_5$ = (3.84 ± 0.27) × 10$^{-11}$ cm$^3$ s$^{-1}$ at 50 Torr, and $k_5$ = (3.95 ± 0.28) × 10$^{-11}$ cm$^3$ s$^{-1}$ at 100 Torr, in good




agreement with the current IUPAC recommended value of $\left(3.70^{+0.45}_{-0.40}\right) \times 10^{-11}$ cm$^3$ s$^{-1}$ (Cox et al., 2020). The data shown in

Figure 6 indicate that the experiment developed in this work is able to measure the spectra and kinetics of reactive species to

a high degree of accuracy and precision, and that a high dynamic range up to at least 20,000 s$^{-1}$ can be achieved.


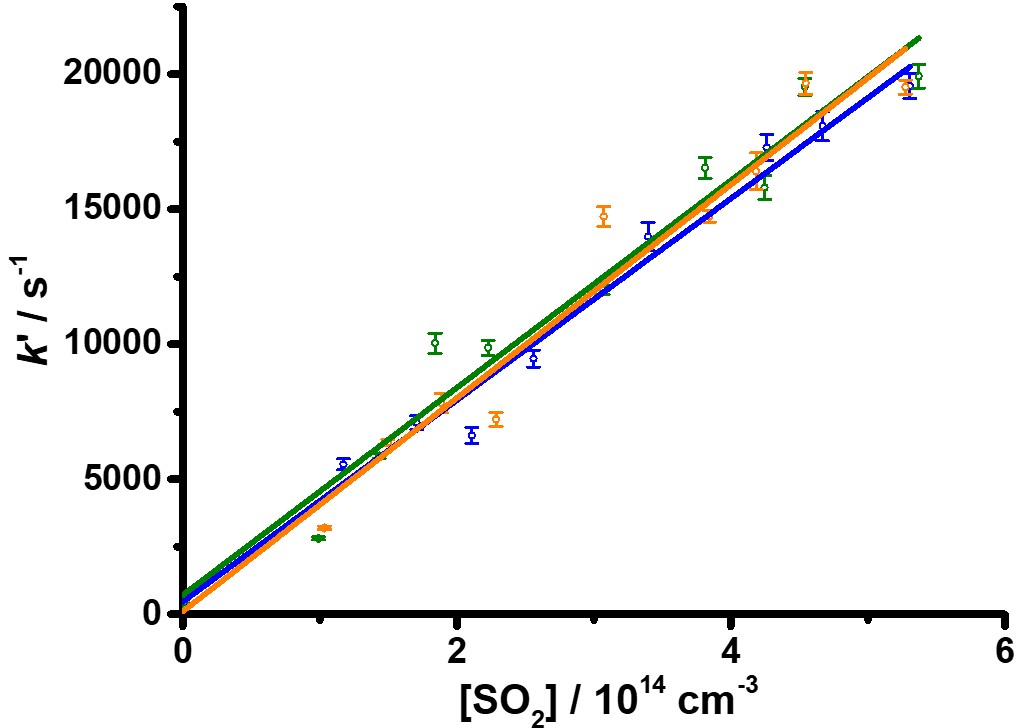

**Figure 6: Bimolecular plots of the pseudo-first-order rate coeffcients, $k'$, determined from fits of Equation 2 to observed CH$_2$OO decays as a function of [SO$_2$] at total pressures of 20 Torr (blue points and line, giving $k_5$ (3.73 ± 0.19) × 10$^{-11}$ cm$^3$ s$^{-1}$, intercept = (500 ± 600 s$^{-1}$)), 50 Torr (green points and line, giving $k_5$ = (3.84 ± 0.27) × 10$^{-11}$ cm$^3$ s$^{-1}$, intercept = (700 ± 900 s$^{-1}$)), and 100 Torr (orange points and line, giving $k_5$ = (3.95 ± 0.28) × 10$^{-11}$ cm$^3$ s$^{-1}$, intercept = (100 ± 900 s$^{-1}$)). The data compare well to the current IUPAC recommendation of $\left(3.70^{+0.45}_{-0.40}\right) \times 10^{-11}$ cm$^3$ s$^{-1}$ (Cox et al., 2020). For these data $T$ = 298 K and initial CH$_2$I$_2$ concentrations were in the range 2.1-9.5 × 10$^{14}$ cm$^{-3}$. Error bars are 1σ.**

Further experiments were performed with the CH$_2$I$_2$/O$_2$/N$_2$/SO$_2$ system to monitor the production of SO$_3$ at ~1388.7 cm$^{-1}$.

Previous work in this laboratory has demonstrated that HCHO is produced in R5 (Stone et al., 2014), and theory has predicted

the co-production of SO$_3$ (Vereecken et al., 2012; Kuwata et al., 2015). Experimental work using step-scan FT-IR spectroscopy

with a resolution of 1 – 4 cm$^{-1}$ have indicated the production of SO$_3$ following photolysis of CH$_2$I$_2$/O$_2$/N$_2$/SO$_2$ mixtures (Wang

et al., 2018), but kinetics of SO$_3$ production have yet to be reported to confirm direct production through R5.





The QCL operating at ~1390 cm$^{-1}$ was tuned to an absorption feature in the SO$_3$ spectrum using a sample of gaseous SO$_3$/N$_2$ in an infrared absorption cell prepared from solid SO$_3$ (Sigma-Aldrich, >99 %) in a glove box purged with N$_2$. For experiments to monitor the kinetics of SO$_3$ production following photolysis of CH$_2$I$_2$/O$_2$/N$_2$/SO$_2$ mixtures, a PicoScope was used to collect the signal owing to lower absorbance signals for SO$_3$ compared to CH$_2$OO for given experimental conditions. The faster sampling rate and memory (5 Gs/s and 12 bit, respectively) of the PicoScope compared to the traditional LeCroy oscilloscope

(sampling rate 1 Gs/s and 8 bit memory) effectively decreases the limit of detection. Factors affecting the limit of detection, and the comparison between the Picoscope and the traditional oscilloscope, are discussed further in Sect. 5.

Figure 7 shows examples of the time-resolved SO$_3$ absorbance, which can be described by a pseudo-first-order growth combined with a first-order loss (Equation 3) and used to provide an alternative determination of $k_5$.

$$A_t = \frac{A_0 k_{\text{growth}}}{(k_{\text{growth}} - k_{\text{loss}})} \left\{ e^{-k_{\text{loss}}t} - e^{-k_{\text{growth}}t} \right\} \qquad \text{(Equation 3)}$$

where $A_t$ is the absorbance at time $t$, $A_0$ is the maximum absorbance which relates to the initial radical concentration and yield of SO$_3$, $k_{\text{growth}}$ is the pseudo-first-order rate coefficient describing the growth of SO$_3$ which is equal to $k_5$[SO$_2$], and $k_{\text{loss}}$ is the first-order rate coefficient describing the loss of SO$_3$ which is expected to be dominated by physical losses such as diffusion out of the probe region.

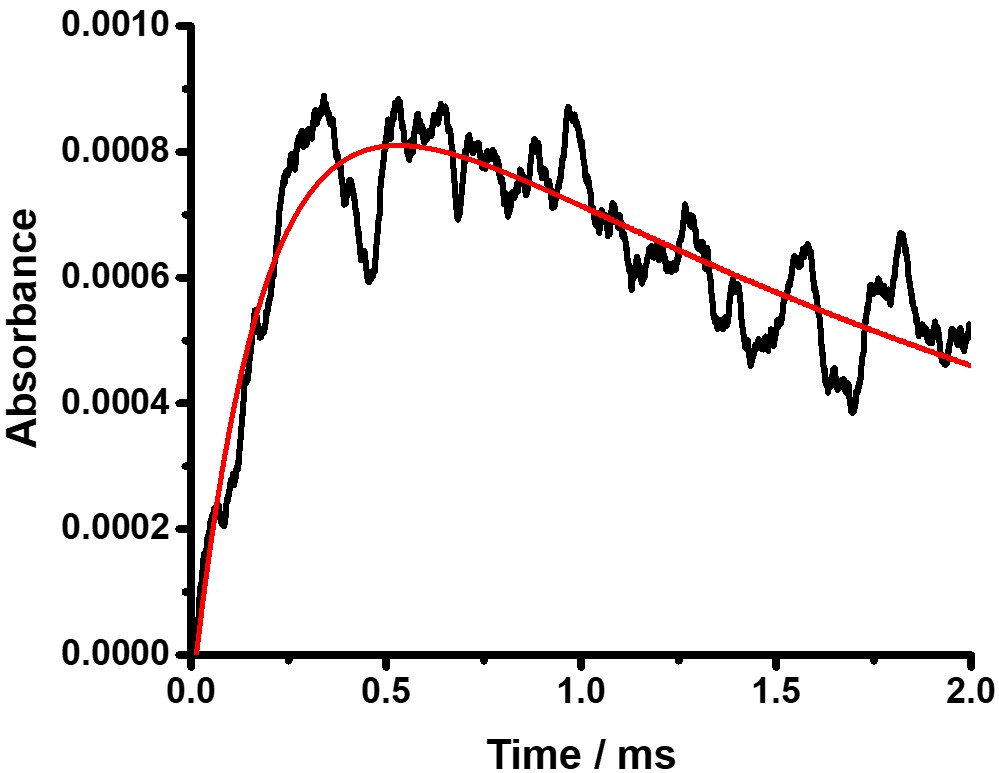


**Figure 7: Time-resolved absorbance signals for SO$_3$ obtained at a wavenumber of 1388.7 cm$^{-1}$. Data are averages of 250 photolysis shots at a total pressure of 50 Torr at 298 K with [SO$_2$] = 2.1 × 10$^{14}$ cm$^{-3}$ and [CH$_2$I$_2$] = 1.6 × 10$^{14}$ cm$^{-3}$. The fit to Equation 3 to determine the kinetics describing the production and loss of SO$_3$ (red) gave $k_{growth}$ = (8250 ± 60) s$^{-1}$ and $k_{loss}$ = (300 ± 10) s$^{-1}$.**

The bimolecular plots obtained through SO$_3$ measurements at total pressures of 20, 50, and 100 Torr are shown in Figure 8 and demonstrate the capability of the instrument to measure the kinetics describing product formation to at least 25,000 s$^{-1}$. The bimolecular plots give $k_5$ = (3.55 ± 0.35) × 10$^{-11}$ cm$^3$ s$^{-1}$ at 20 Torr, (3.87 ± 0.28) × 10$^{-11}$ cm$^3$ s$^{-1}$ at 50 Torr, and (4.05 ± 0.19) × 10$^{-11}$ cm$^3$ s$^{-1}$ at 100 Torr, in excellent agreement with that determined through measurements of CH$_2$OO described above and the current IUPAC recommendation (Cox et al., 2020). These results thus demonstrate direct production of SO$_3$

through R5, and the capability of the experiment to identify and monitor reaction products.



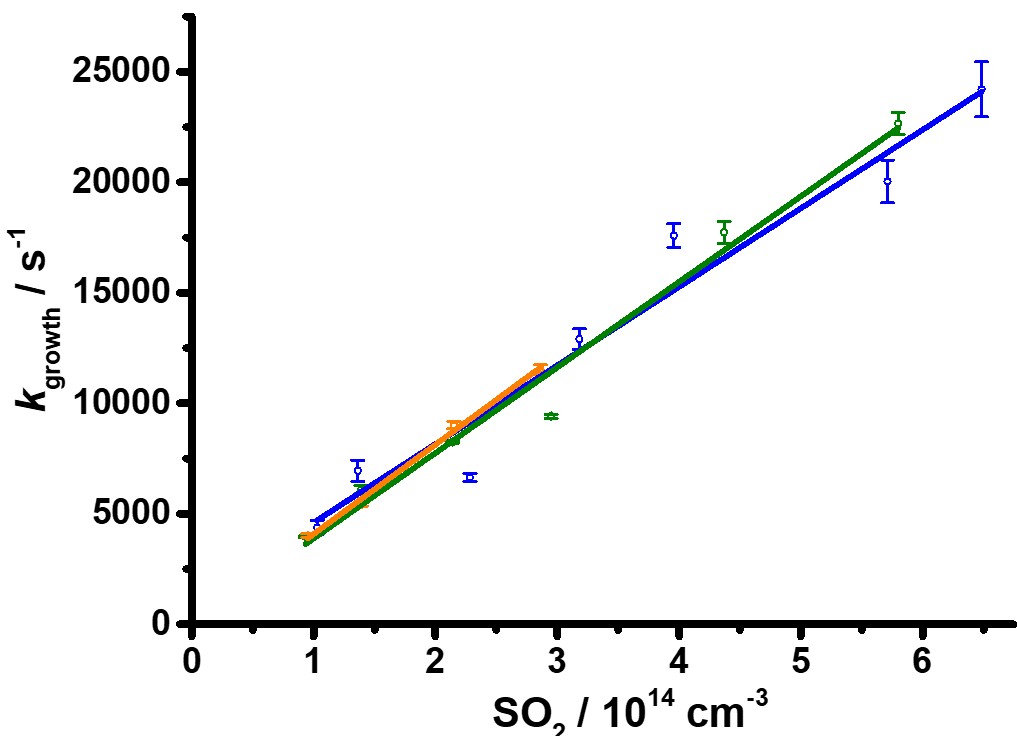

**Figure 8: Bimolecular plots of the pseudo-first-order rate coefficents, $k'$, determined from the observed $SO_3$ profiles as a function of [$SO_2$]. Data were obtained at total pressure of 20 Torr (blue points and line, giving $k_5 = (3.55 \pm 0.35) \times 10^{-11}$ cm$^3$ s$^{-1}$, intercept = (1000 ± 1400 s$^{-1}$)), 50 Torr (green points and line, giving $k_5 = (3.87 \pm 0.28) \times 10^{-11}$ cm$^3$ s$^{-1}$, intercept = (100 ± 900 s$^{-1}$)), and 100 Torr (orange points and line, giving $k_5 = (4.05 \pm 0.19) \times 10^{-11}$ cm$^3$ s$^{-1}$, intercept = (400 ± 400 s$^{-1}$)). The data compare well to the current IUPAC recommendation of $\left(3.70^{+0.45}_{-0.40}\right) \times 10^{-11}$ cm$^3$ s$^{-1}$ (Cox et al., 2020). Error bars are 1σ.**

## 5 Limit of detection

The limit of detection can be determined from the variability of the baseline absorbance, (i.e. in the absence of any absorbing species), which should be equal to zero and for which deviations from zero are determined only by noise. In order to detect an absorbance signal above the baseline, the signal must be greater than the noise (i.e. the signal-to-noise ratio must be greater than one) and the limit of detection can thus be defined as the standard deviation of the noise. Figure 9 shows how the 1σ limit of detection varies with the number of samples, which for measurements involving reactive species (Section 4) is equal to the number of photolysis shots and is determined for the baseline given for the pre-photolysis period (-4000 μs to -500 μs, where $t = 0$ is the time at which the photolysis laser is fired).

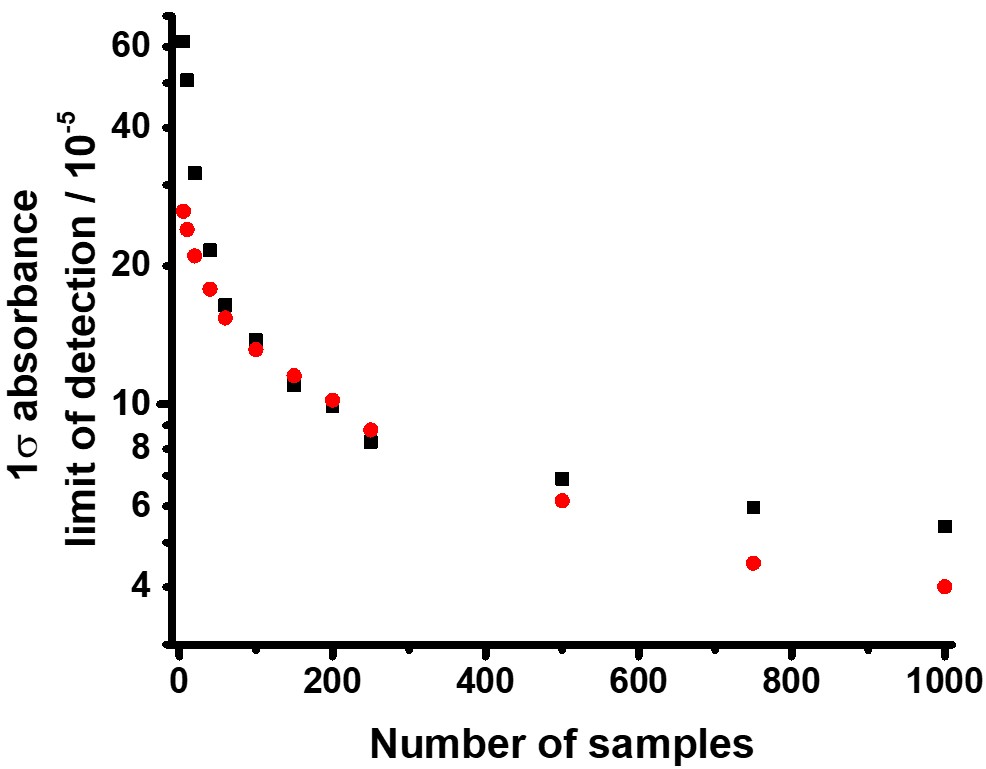

**Figure 9: Variation of the 1σ limit of detection with the number of samples, which is equivalent to the number of photolysis shots for measurements involving reactive species. The 1σ limit of detection is defined as the standard deviation of the baseline absorbance,**
**and is given for the time period equivalent to the pre-photolysis period (-4000 μs to -500 μs, where $t = 0$ is the time at which the photolysis laser is fired) for measurements of reactive species. Measurements are shown for the traditional oscilloscope (LeCroy Waverunner-2, LT262, 350 MHz, 1 GS/s sample rate, 8 bit resolution) in black and the PicoScope (Pico Technology, PicoScope 6402C, 250 MHz, 5 GS/s sample rate, 12 bit resolution) in red.**

The limit of detection is given for measurements with the traditional oscilloscope and the PicoScope, with some improvement

to the limit of detection achieved on using the PicoScope, owing to greater memory and sampling rate (12 bit and 5 Gs/s)

compared to the traditional oscilloscope (8 bit and 1 G/s), which effectively increases the number of measurement points within

a sample within a given time period. For the traditional oscilloscope, a limit of detection of $8.3 \times 10^{-5}$, in absorbance terms, is

achieved for 250 samples, which is reduced to $5.4 \times 10^{-5}$ for 1000 samples. For the estimated path length of $(218 \pm 20)$ cm and

typical IR absorption cross-sections of ${\sim}10^{-19}$ cm$^2$, the limit of detection for these data in terms of concentration can thus be

estimated as ${\sim}3.8 \times 10^{12}$ cm$^{-3}$ for 250 samples and ${\sim}2.5 \times 10^{12}$ cm$^{-3}$ for 1000 samples, which compares well with alternative

IR-based techniques (Taatjes and Hershberger, 2001; Roberts et al., 2020). For measurements of $CH_2OO$, which has relatively


high IR absorption cross-sections on the order of $10^{-18}$ cm$^2$ at ~1286 cm$^{-1}$, the limit of detection for 250 samples using the traditional oscilloscope is thus ~$2.5 \times 10^{11}$ cm$^{-3}$. For the PicoScope, the limit of detection is significantly better than that for

the traditional oscilloscope when the number of samples is low, with the limits of detection becoming more comparable as the number of samples is increased. For 1000 measurements, the PicoScope gives a limit of detection of $4.0 \times 10^{-5}$ in absorbance terms, which for species with IR cross-sections on the order of ~$10^{-19}$ cm$^2$ gives a limit of detection of ~$1.8 \times 10^{12}$ cm$^{-3}$ in terms of concentration. For CH$_2$OO, with relatively high IR cross-sections of ~$10^{-18}$ cm$^2$ at ~1286 cm$^{-1}$, the limit of detection is thus on the order of ~$1.8 \times 10^{11}$ cm$^{-3}$.

**6 Conclusions and future improvements**

This work presents the characterisation and initial experiments performed using a new instrument based on mid-infrared QCL absorption spectroscopy to investigate the chemistry of reactive species with high spectral and temporal resolution. We have presented details of the experimental setup (Section 2), results obtained for time-averaged measurements of stable species (Section 3), and those for time-resolved measurements of reactive species (Sect 4).


We have demonstrated the application of the instrument to measurements of the IR spectra of reactive species which can be used to identify reactive species and their reaction products, as well as to monitor reaction kinetics. The capabilities of the instrument have been demonstrated through measurements of the $v_4$ band of the infrared spectrum of the CH$_2$OO Criegee intermediate, produced by laser flash photolysis of CH$_2$I$_2$/O$_2$/N$_2$ gas mixtures at $\lambda = 266$ nm, and through measurements of the

kinetics of the reaction between CH$_2$OO and SO$_2$ under a range of conditions. The results have demonstrated the ability to measure reaction kinetics through monitoring of either reactants or reaction products, with the potential for the identification of reaction products and measurements of product yields. Results have shown that SO$_3$ is a reaction product in the reaction of CH$_2$OO with SO$_2$, with preliminary results indicating that there is no pressure dependence in the yield of SO$_3$.

The instrument described in this work has applications in atmospheric chemistry and chemical kinetics, with wider potential uses in trace gas analysis in industrial processes and medical diagnostics. Future development of the instrument will focus on improvements in the signal-to-noise ratio and limit of detection. Improvements to the limit of detection could be achieved through further increases to the sampling rate and through the use of a balanced detector or lock-in amplification techniques to improve the signal-to-noise ratio. Additional improvements to the limit of detection in concentration terms are achievable

through the use of a mirror arrangement for the probe beam in which the optics are internal to the reaction cell, thereby reducing intensity losses as the probe beam passes through the cell windows on each pass through the cell and enabling significant increases to the effective pathlength.



**Acknowledgements**

The authors would like to thank the Natural Environment Research Council (NERC) for funding (grant references
NE/L010798/1 and NE/P012876/1) and for the award of a studentship as part of the SPHERES DTP scheme.

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
