# Peer review of "Identification, monitoring, and reaction kinetics of reactive trace species using time-resolved mid-infrared quantum cascade laser absorption spectroscopy: Development, characterisation, and initial results for the Criegee intermediate CH2OO"

_Atmospheric Measurement Techniques, 2021_

## Author Comment (AC2)

We would like to thank the reviewers for their time taken to review the manuscript and for their helpful comments which will improve the manuscript.

**RC1**

The paper describes a new set-up combining laser-photolysis with time-resolved absorption spectroscopy using cw-QCL lasers. For validation, a part of the spectrum of the simplest Criegee intermediate, $CH_2OO$, has been measured as well as the rate constant of the reaction between $CH_2OO$ and $SO_2$. The paper shows the future potential of the experimental set-up, without showing any new data. I have a few minor remarks that could improve the paper.

Figure 3: it would be interesting to see the signal on a longer time scale to have an idea about the influence of diffusion.

We have replaced the plot in the manuscript with one showing the full time series, which was measured between -1.5 ms and 3 ms and so shows limited impacts of diffusion.

Figure 4: I guess $CH_2I_2$ does not absorb in this wavelength range, because otherwise it would influence the measured spectrum?

Yes, we do not observe any effects of $CH_2I_2$ absorption in this region and the HITRAN database indicates there is no appreciable absorption by $CH_2I_2$ ~1286 cm$^{-1}$. We have added a comment on this to the caption.

Even though possible to calculate by everybody, it would be good to indicate the time it took to measure the spectrum to give an idea to the reader. I calculated that it has taken more than 60 hours to measure the spectrum, which is rather long given the quality of the obtained spectrum compared to the spectrum available in the literature. Why is the quality so low? Given the S/N ratio of the kinetic decay in Figure 5, which has been obtained also from averaging over 1000 laser pulses and with even lower concentration of $CH_2I_2$, I would have expected a much better S/N ratio in the spectrum.

The reviewer is correct that the method used in this paper required a lengthy series of experiments. We agree that the S/N ratio is lower than that achieved in the previous work, but we dispute that the quality of the spectrum is low. The high $CH_2OO$ reactivity and concentrations on the order of 10$^{12}$ cm$^{-3}$ make high resolution spectroscopy challenging. For the experiments reported in this work, the spectrum was obtained using the temporal profiles recorded at each wavenumber, rather than an FT-IR method whereby the whole spectrum is obtained for each photolysis shot. Slight variations in the photolysis laser fluence from shot to shot in this work thus lead to slight differences in the initial $CH_2OO$ concentration between experiments which impacts the S/N for the spectrum. For future work we intend to measure the laser fluence for each photolysis shot so that results can be normalised for this.

We also note that the technique used in this work is not proposed for detailed spectroscopic studies, but rather for the identification of species to be monitored in kinetics experiments. The full band for $CH_2OO$ is included in this work to provide confidence in the capability of the technique, but in most cases only a partial band will be required for unambiguous identification.

Line 274: I don't understand the sentence: "The pre-photolysis region was defined as -4000 μs to -500 μs, owing to detection of some radiofrequency noise associated with the Q-switch delay of the photolysis laser which was set to 280 μs, and the post-photolysis region as 500 μs to 6000 μs, where t = 0 is the time at which the photolysis laser is fired." Is the Q-switch at -280 μs, and does the noise still influence even after the laser pulse, or why is there such a long post-photolysis delay? The signal in Figure 5 looks perfect from the first μs on. Maybe show an example of a typical signal with pre-trigger to clarify?

Yes, the Q-switch delay of 280 μs results in the Q-switch firing 280 μs before the 266 nm pulse (i.e. at -280 μs). We have clarified this in the main text as follows:

"The pre-photolysis region was defined as -4000 μs to -500 μs, owing to detection of some radiofrequency noise associated with the Q-switch delay of the photolysis laser which was set to 280 μs (i.e. the Q-switch fires at $t$ = -280 μs), and the post-photolysis region as 500 μs to 6000 μs, where $t$ = 0 is the time at which the photolysis laser is fired."

The radiofrequency noise generated by the Q-switch has a small impact on the measurement for a period that lasts beyond $t = 0$. In order to avoid this completely, we analyse data for the spectral measurements from $t = 500\ \mu s$ onwards, although using data from t = 0 would make only a small difference to the overall results.

Why do the data in Figure 5 show that the diode does not drift? Because Figure 5 is the average of 1000 photolysis shots, the average could still be a good quality decay, even if the wavelength changed during the measurement, no?

This is a good point, and should have referred more clearly to repeated measurements taken in separate experiments rather than the averaged measurements shown in Figure 5. We have re-worded the text as follows:

"The QCL can be tuned to a particular spectral feature and can remain tuned to that feature for prolonged periods of time during which kinetics experiments, and repeat measurements, can be performed without any drift in spectral position."

It would make it easier for the reader if Table 1 would also contain the corresponding reaction numbers.

We have added these to the table and modified the text at the start of Section 4.2 to include loss by diffusion labelled as R6.

The difference in pathlength is somewhat strange: would it not have been possible to measure both kinetics with the same alignment to verify if this is the reason for the change? Also I'm wondering if the $k_{phys}$ has changed a lot between both experiments: with $CH_2OO$ it is 500 s$^{-1}$, which should leave to a visible increase of the $CH_3I$ kinetic in Figure 3, but Figure 3 looks like a $k_{phys}$ well below 100 s$^{-1}$.

The difference in path length is relatively small and likely arises from small changes in alignment, or potential uncertainties in the absorption cross-sections or concentrations of $CH_3I$ or $CH_2OO$ used in the analysis. Unfortunately, there was a significant time period between the measurements involving $CH_3I$ and those involving $CH_2OO$ where access to the laboratory was restricted, with the result that the system had to be realigned between the two sets of experiments. The realignment also likely impacted $k_{phys}$, since this represents the diffusion of the species under investigation out of the probe region, and the extent of diffusion may depend, in part, on the overlap between the photolysis and probe lasers. However, we would like to note that for measurements of first-order and pseudo-first-order kinetics, knowledge of the path length is not required and is primarily included in this work to demonstrate the sensitivity of the instrument.

The error bars in Figure 6 are very small: I guess they are statistical from the exponential fitting? Please show a few examples of decays at the highest pseudo-first order rates, it would be interesting to see the quality of the data, especially with respect to the noise from the photolysis laser described above. Maybe this could be done as supplementary material.

Yes, the uncertainties are the statistical uncertainties obtained from the fits to the exponential decays. We have added comments to the captions to Figures 6 and 8 to clarify this:

"Error bars are 1σ from the fits to Equation 2/3".

We have also added some more example decays as an inset to Figure 6.

Because you can know the initial $CH_2OO$ concentration as well as the formed $SO_3$, you should be able to measure the yield of $SO_3$. Did you try?

Measurement of the $SO_3$ yield requires knowledge of the $SO_3$ absorption cross-section at the spectral position used in these experiments, which is subject to large uncertainties, and so we did not attempt to characterise the yield in this work but this should be possible with the instrument described in the manuscript.

---

## Author Comment (AC3)

We would like to thank the reviewers for their time taken to review the manuscript and for their helpful comments which will improve the manuscript.

**RC2**

Mir et al. report the IR-spectroscopy and time-resolved detection of the simplest Criegee intermediate,  $CH_2OO$ , and the time-resolved detection of  $SO_3$  from the reaction of  $CH_2OO$  with  $SO_2$ . These measurements have been made using a new apparatus involving a mid-IR quantum cascade laser as a tunable IR source, which enables the detection of multiple species. The new results are in accord with previous measurements in the literature, supporting the reliability of the new apparatus. This is a nice study, which is within the scope of this journal. The paper is well written, the literature appropriately cited, and the methods and analysis clearly stated. I anticipate that this new apparatus will provide novel kinetic and mechanistic insights to atmospherically important reactions. I have only a few minor comments and suggestions, detailed below.

Fig 3: you determine that the expected change in CH3I concentration on photolysis under your experimental conditions is 4 %. It would be good to also state the percentage change that you measure experimentally.

Evaluation of the observed percentage change in  $CH_3I$  concentration requires knowledge of the  $CH_3I$  absorption crosssection at the probe wavelength and the effective path length, the determination of which relied on the calculation of a 4 % change in concentration based on the laser fluence and absorption cross-section of  $CH_3I$  at the photolysis wavelength. Assessment of the observed percentage change in  $CH_3I$  concentration is therefore somewhat circular and inevitably agrees with the calculated value of 4 %.

If I understand this correctly, noise from the Q-switch inhibits reliable measurements from -500 to+500  $\mu$ s. It would be beneficial to the reader to show an example of the full kinetic trace (including this time window) in the supplementary material.

Figure 3 uses data recorded from -1.5 ms to 3 ms and defines the pre-photolysis region from -1.5 ms to -0.5 ms. We have clarified the definition of the pre-photolysis region for these experiments in the caption to the figure. We have also replaced the figure in the manuscript to show the full time series.

**What vapor pressure of $CH_2I_2$ was used in the calculation of $[CH_2I_2]$ ?**

The vapour pressure was estimated as 0.2 Torr from the vapour pressure at room temperature (1.2 Torr) and the standard enthalpy of vaporisation (45.6 kJ mol-1). We note that the vapour pressure is uncertain since the temperature inside the bubbler is not known exactly, but the  $CH_2I_2$  concentration was estimated from separate previous experiments under identical bubbler and flow conditions in which the  $CH_2I_2$  concentrations were measured by UV absorption spectroscopy, and so the estimated  $CH_2I_2$  concentration ought to be robust. We also note that the absolute concentration of  $CH_2I_2$  is not required in the analysis of any results obtained in this work.

Figs 2 and 4: I think the vertical inversion of the literature spectra make it easier for the reader to see the features of the new spectra, but it is difficult to compare the relative intensities of spectral features in the literature vs. current spectra in these plots. I suggest that in the supplementary material, the literature and current spectra are overlaid so this comparison can be more easily made.

For  $SO_2$  (Figure 2), the number of peaks in the absorption spectrum makes it difficult to compare the spectrum measured in this work and that given on the HITRAN database when the two are overlaid. We have included the comparison below to demonstrate this, but prefer not to include this figure in the manuscript or supplementary information as we do not feel it helps to compare the two spectra.

Normalised observed spectrum (black) for SO2 with the corresponding normalised spectrum reported on the HITRAN database (red).

For CH2OO (Figure 4), we have added the suggested plot as an inset to the figure in the manuscript.

Fig 5: Why is the pre-photolysis signal (at least that before the Q-switch noise at -500 µs) not shown ?

We have extended the scale to show the pre-photolysis signal on the plot.

I agree with reviewer #2 that it would be beneficial to show some additional examples of  $CH_2OO$  decays in the supplementary material.

We have added some additional examples as an inset to Figure 6.